# ADAPTING SELF-SUPERVISED REPRESENTATIONS AS A LATENT SPACE FOR EFFICIENT GENERATION

**Ming Gui**[1,2]* **Johannes Schusterbauer**[1,2]* **Timy Phan**[1,2]

**Felix Krause**[1,2] **Josh Susskind**[3] **Miguel Angel Bautista**[3] **Björn Ommer**[1,2]

[1] CompVis @ LMU Munich   [2] Munich Center for Machine Learning (MCML)   [3] Apple

## ABSTRACT

We introduce **Rep**resentation **Tok**enizer (RepTok), a generative modeling framework that represents an image using a single continuous latent token obtained from self-supervised vision transformers. Building on a pre-trained SSL encoder, we fine-tune only the semantic token embedding and pair it with a generative decoder trained jointly using a standard flow matching objective. This adaptation enriches the token with low-level, reconstruction-relevant details, enabling faithful image reconstruction. To preserve the favorable geometry of the original SSL space, we add a cosine-similarity loss that regularizes the adapted token, ensuring the latent space remains smooth and suitable for generation. Our single-token formulation resolves the spatial redundancies of the 2D latent space and significantly reduces training costs. Despite its simplicity and efficiency, RepTok achieves competitive results on class-conditional ImageNet generation and extends naturally to text-to-image synthesis, reaching competitive zero-shot performance on MS-COCO under extremely limited training budgets. Our findings highlight the potential of fine-tuned SSL representations as compact and effective latent spaces for efficient generative modeling. We release our code at `https://github.com/CompVis/RepTok`.

## 1 INTRODUCTION

In recent years, diffusion- (Ho et al., 2020; Kingma et al., 2021; Song & Ermon, 2019) and flow-based (Lipman et al., 2023; Liu et al., 2023b; Ma et al., 2024) models have emerged as powerful generative modeling frameworks, capable of synthesizing high-quality images (Ramesh et al., 2022; Rombach et al., 2022; Dhariwal & Nichol, 2021) and videos (Ho et al., 2022). However, these models typically come with substantial computational demands since they regress vector fields in the high-dimensional pixel space of images. Latent Diffusion Models (Rombach et al., 2022) address this challenge by decomposing the generative modeling task into two stages. By first compressing images into a lower-dimensional latent space via a pre-trained Variational Autoencoder (Kingma et al., 2013), LDMs abstract away imperceptible details, enabling the generation process to solely focus on semantic content and drastically reducing computational costs during training and inference (Esser et al., 2021; 2024; Fuest et al., 2024; Schusterbauer et al., 2024). However, despite these computational advantages, the latent space is still organized in a two-dimensional grid structure, which fails to exploit the high spatial redundancies inherent to natural images.

Recent efforts have sought to improve latent generative paradigms along two directions. *TiTok* (Yu et al., 2024a) tries to exploit spatial redundancies and replaces the default 2D spatial grid in latent diffusion with a transformer-based encoder–decoder that represents images as 1D latent sequences, achieving compact encodings with as few as 32 discrete tokens. In parallel, *REPA* (Yu et al., 2024b) leverages the rich representations of pre-trained self-supervised learning (SSL) models to accelerate the convergence of latent diffusion models, by distilling the semantic knowledge into the diffusion model via a cosine similarity loss between their respective feature representations.

---

*Equal Contribution

†Experimentation, including use of pre-trained models, were completed by university collaborators.

Figure 1: Comparison of our single-token MLP-Mixer generator against transformer-based baselines (DiT, SiT), as well as representation-aligned models like REPA. RepTok attains competitive generative performance while reducing training cost by over 90% owing to its compact latent space and lightweight architecture. All results reported without CFG. For fair comparison, we employ an encoder and decoder trained on general-domain data.

In this work, we extend these two directions by exploring more powerful uses of SSL representations. While REPA accelerates training primarily through feature alignment on the 2D spatial grid, we demonstrate that self-supervised models can be leveraged more directly: with minimal but crucial fine-tuning, pooled 1D SSL representations themselves constitute effective latent spaces for generative modeling. These representations exhibit smooth, semantically structured geometry that is well-suited for generation, while simultaneously eliminating the spatial redundancies inherent in 2D grid-based latents. Specifically, we show that the pooled 1D output from the `[cls]` token alone provides a compact yet expressive representation that not only captures high-level semantics but also preserves sufficient spatial detail to enable high-fidelity reconstruction.

Our **Rep**resentation **Tok**enization approach, termed RepTok, builds on a pre-trained SSL encoder that is lightly fine-tuned and trained jointly with a generative decoder. We train the decoder with a standard flow matching objective, complemented by a cosine-similarity loss that regularizes the latent representation to remain close to its original smooth and semantically structured space, which is well-suited for generation. Without auxiliary perceptual (Zhang et al., 2018) or adversarial (Esser et al., 2021) losses, the resulting model is able to faithfully decode the single-token latent representation into the pixel space. This design enables highly efficient image synthesis training, allowing us to use simple, attention-free architectures such as MLP-Mixers (Tolstikhin et al., 2021) for accelerated ImageNet training (see Figure 1). Furthermore, we show that the framework naturally extends to text-to-image (T2I) synthesis: by incorporating cross-attention to integrate textual conditioning, our model achieves competitive zero-shot performance on the COCO (Lin et al., 2014) benchmark under an extremely constrained training budget (see Figure 7). We state our contributions as follows:

- We show that self-supervised vision transformers can be used more powerfully than just guiding generative training: with minimal adaptation of the semantic token, their smooth and semantically structured latent spaces can directly act as encoders for generative modeling. By injecting the necessary fine-grained information into this semantic token, we enable faithful reconstruction while simultaneously eliminating the spatial redundancies inherent in 2D grid-based latents. Coupled with a generative decoder, this setup allows accurate image reconstruction from a single continuous token.

- Exploiting this autoencoder design, we introduce a lightweight and optionally attention-free pipeline for latent generative modeling. This drastically reduces training compute while preserving quality, achieving competitive ImageNet generation at a fraction of the cost of transformer-based diffusion baselines.

- We show that RepTok scales effectively to text-to-image synthesis, achieving competitive zero-shot results on MS-COCO with under 20 hours of training on four A100 GPUs.

## 2 RELATED WORK

**Latent space generation** Early approaches such as PixelVAE and VQVAE(Gulrajani et al., 2016; Razavi et al., 2019; Van Den Oord et al., 2017) demonstrated that generative modeling within compact latent spaces significantly improves sampling quality and efficiency. VQGAN (Esser et al., 2021) integrates vector-quantized variational autoencoders with adversarial losses to construct discrete latent codebooks. Subsequently, these discrete tokens are leveraged by autoregressive transformers

Figure 2: **Overview of our pipeline**. (a) Joint fine-tuning of the `[cls]` token of SSL encoder $\mathcal{E}$ and training of the generative decoder $\mathcal{D}$ for image reconstruction. (b) Training of the generation model $\mathcal{G}$ to synthesize frozen encoder outputs, which constitute the latent space $z = \mathcal{E}(x)$. (c) Inference pipeline, where the latent space $z$ is first generated and subsequently decoded into the pixel space.

for image generation tasks. Latent Diffusion Models (LDMs) (Rombach et al., 2022) brought this concept into the diffusion models, operating in learned spatial latent spaces that preserve semantic content and abstract away perceptual detail. This approach has since become foundational across modalities including images (Peebles & Xie, 2023; Ma et al., 2024; Pernias et al., 2024), audio (Liu et al., 2023a), and video (Ho et al., 2022; Blattmann et al., 2023b;a; Kong et al., 2024).

**Pre-trained representations in diffusion models**    Leveraging pre-trained representations has been shown to improve image generation Kouzelis et al. (2025); Yao et al. (2025). REPA (Yu et al., 2024b; Wang et al., 2025) accelerates diffusion training by aligning diffusion features with DINO embeddings. Closely related to our approach is RCG (Li et al., 2024), which employs a two-stage pipeline: first generating a predefined semantic representation and then transporting it to the pixel space. However, RCG primarily targets unconditional synthesis and thus leaves the representation space unchanged. In contrast, our objective is faithful reconstruction and generation, similar to the role of the latent space in VAEs. This requires not only semantic but also low-level visual information. We address this by injecting fine-grained details into the representation space, enabling both faithful reconstruction and high generative performance. Concurrent works like RAE (Zheng et al., 2025) and SVG (Shi et al., 2025) use the full spatial grid of SSL features, thus operating in a high-dimensional structured latent space. Our approach is fundamentally different: RepTok relies solely on the pooled semantic token, discarding all spatial tokens and learning to represent an image with a single compact vector. This yields a much more aggressive compression. Additionally, a key difference to SVG whose objective is aligning to the SSL representation, we directly employ the pooled semantic output as the latent representation itself.

**Global information latent spaces**    Recent work has explored 1D tokenization beyond spatial grid latents. TiTok (Yu et al., 2024a) encodes images into compact sequences of as few as 32 discrete tokens with a ViT encoder, enabling efficient autoregressive generation. ElasticTok (Yan et al., 2024) extends this idea with adaptive token counts per frame, while FlexTok (Bachmann et al., 2025) introduces variable-length ordered tokens for coarse-to-fine generation. Our approach differs in the following key aspects: First, we operate in a continuous latent space, avoiding quantization and enabling fully differentiable diffusion training. Second, we directly exploit the `[cls]` token of SSL vision transformers as a compact latent, yielding smooth and semantically structured manifolds. Unlike discrete tokenizers, Diffusion Autoencoders (Preechakul et al., 2022) extract semantic information into a continuous latent space and utilize a jointly trained diffusion model for reconstruction. As the latent space is mostly semantic, image reconstruction requires an additional subcode $x_T$, obtained by mapping the image back to the Gaussian noise space using conditional DDIM sampling (Song et al., 2020). By contrast, our method reconstructs images faithfully from a single latent $z$ alone. A concurrent work, AToken (Lu et al., 2025), proposes a unified visual tokenizer designed to operate consistently across multiple modalities.

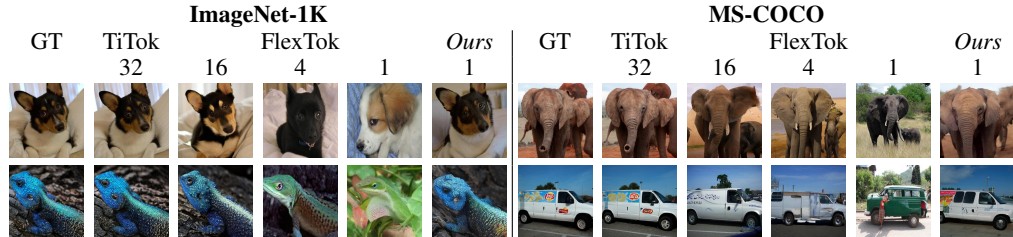

Figure 3: We introduce *RepTok*, a compact visual tokenizer that builds upon pre-trained SSL representations. Our approach augments these representations with additional necessary information to enable images to be faithfully encoded as a single continuous token, which allows for both high-fidelity image reconstruction and synthesis. The third row indicates the number of tokens for reconstruction.

# 3 METHOD

## 3.1 PRELIMINARIES

**Flow Matching** models learn vector fields that map between two terminal distributions: $p(x_0)$, typically a simple prior distribution such as a standard Gaussian distribution, and $p(x_1)$, the target data distribution. Let $\mathbb{R}^d$ be the space that $x_0$ and $x_1$ reside in, and let $v_\theta(t, x)$ represent the time-dependent vector field to be learned with $t \in [0, 1]$. The underlying dynamics of flow matching models are then governed by the ordinary differential equation (ODE) $dx = v_\theta(x, t)$. A common choice for the interpolant between $x_0$ and $x_1$ is the linear interpolant (Liu et al., 2023b), defined as $x_t = tx_1 + (1 - t)x_0$. The vector field $v_\theta$ can then be optimized using the following training objective with a randomly sampled $t$ and the corresponding $x_t$ (Lipman et al., 2023; Liu et al., 2023b; Schusterbauer et al., 2025):

$$\mathcal{L} = \mathbb{E}_{t, x_0, x_1} ||v_\theta(x_t, t) - (x_1 - x_0)||. \tag{1}$$

To sample from a flow matching model, one simply integrates along the trajectory defined by the learned ODE. This can be accomplished using numerical integration techniques such as the forward Euler method, with the update rule given by $x_{t+t_\Delta} = x_t + t_\Delta \mathbf{v}_\theta(x_t, t)$, where $\forall t \in [0, 1), t_\Delta = 1/N$, and $N$ being the number of function evaluations (NFE).

## 3.2 REPTOK: REPRESENTING IMAGES AS A SINGLE TOKEN

TiTok (Yu et al., 2024a) represents a significant advancement over traditional VAEs by overcoming their inherent 2D tokenization grid constraints. Unlike conventional approaches, where each token is restricted to attending only to a fixed image grid, TiTok enables tokens $z$ to attend freely to the entire image. However, despite these improvements, TiTok typically still relies on multiple tokens to effectively encode an image. In this work, we show that continuous latent spaces can achieve even greater efficiency in few-token regimes. Specifically, we demonstrate that a single continuous token, derived from a pre-trained encoder, can be used together with a generative decoder to synthesize high-fidelity reconstructions.

**Finetuned Self-supervised Models are Faithful Encoders** It is well established that models such as CLIP (Radford et al., 2021), MAE (He et al., 2022) and DINO (Caron et al., 2021; Oquab et al., 2024) models encode highly informative representations and demonstrate a strong understanding of images, as evidenced by their effectiveness in various downstream tasks, including image classification (Radford et al., 2021; Caron et al., 2021; Oquab et al., 2024) and semantic segmentation (Zhang et al., 2023). This capability is further demonstrated by the existence of unCLIP models (Ramesh et al., 2022; Rombach et al., 2022), which can generate image variations from noise using only a single CLIP embedding. While this observation confirms that generative models can synthesize images from extremely compact bottlenecks (for unCLIP (Ramesh et al., 2022) $z \in \mathbb{R}^{1 \times 512}$), we hypothesize that the variations of the outputs arise from the fact that CLIP models are not explicitly trained to preserve exact pixel locations but instead optimize a contrastive loss with corresponding textual descriptions, thereby capturing only high-level semantic features.

Motivated by these observations, we explore and unlock the potential of leveraging a pretrained encoder $\mathcal{E}$ that already possesses a comprehensive understanding of image content. To this end, we

Image A ⟵————— Interpolation —————⟶ Image B

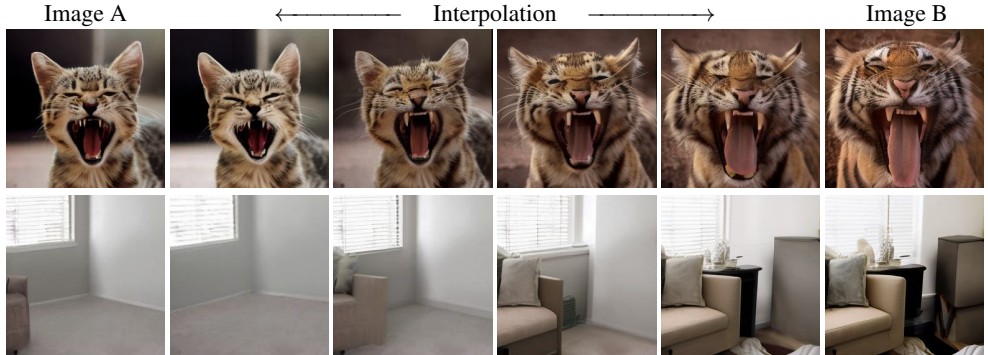

Figure 4: **Latent space interpolation.** We observe smooth transitions not only in semantic content but also in spatial configuration. This indicates that our method successfully integrates low-level spatial information while preserving the properties of the pretrained encoder's latent space, and facilitates generation within the learned representation. We provide more samples in the Appendix.

introduce a novel training strategy that leverages a pretrained self-supervised learning (SSL) model with a transformer-based architecture as the encoder. These models typically incorporate a class token (typically referred to as the `[cls]` token) that is trained, either explicitly or implicitly, to aggregate information from image patches. However, such pretrained models are often optimized for downstream tasks and may consequently, as an example, *underrepresent* low-level visual details critical for image reconstruction. To address this limitation, we propose a targeted adaptation strategy that *only* updates the class token embedding while keeping the remainder of the encoder frozen. Remarkably, we find that this minimal intervention is sufficient to inject the necessary visual detail into the representation. Empirical results reveal that with only the class token being fine-tuned, the system is capable of producing reconstructions with high fidelity across a range of SSL backbones including DINOv2 (Oquab et al., 2024), MAE (He et al., 2022) and CLIP (Radford et al., 2021). We demonstrate our reconstructions in Figure 3.

**Training the Encoder together with a Generative Decoder**   While the SSL-pretrained encoder $\mathcal{E}$ remains largely frozen, a supervisory signal is still required to inject reconstruction-relevant information into the class token. Additionally, a decoder is necessary to map the resulting single-token latent representation back into pixel space. To this end, we jointly train the encoder $\mathcal{E}$ and a generative decoder $\mathcal{D}$ in a continuous manner, using a simple but effective flow matching loss.

The generative decoder $\mathcal{D}$ is trained end-to-end alongside the encoder $\mathcal{E}$ to learn a mapping from randomly sampled Gaussian noise $\epsilon$ to the target image $x$. We follow principles similar to the conditioning mechanism employed in MMDiT (Esser et al., 2024) and concatenate the latent token $z = \mathcal{E}(x)$ with the noisy image tokens. The resulting training objective is formulated as a flow matching loss as in Equation (1), which optimizes both the encoder and the decoder:

$$\mathcal{L} = \mathbb{E}_{t,x_0,x_1}||v_\theta(t, x_t, z) - (x_1 - x_0)||. \tag{2}$$

To improve computational efficiency and remain consistent with the SiT framework (Ma et al., 2024), we adopt a pretrained 2D VAE (Rombach et al., 2022) so that the generative decoding process operates within a learned latent space rather than directly in pixel space.

**Cosine-Similarity Loss**   We observe that the `[cls]` tokens of self-supervised vision encoders already provide a smooth, semantically structured space. Hence, our goal during training is to maintain this well-regularized space while still allowing the token to integrate the fine-grained information the decoder needs for faithful reconstructions. Freezing the `[cls]` token leads to poor reconstruction quality, as indicated in Figure 5. Conversely, leaving the encoder completely unconstrained pulls the token far away from the well-regularized space, removing the prerequisite for later generative modeling. We find that only unfreezing the `[cls]` while fixing all other encoder weights strikes a good balance between integration of more information and maintaining the original regularization. To constrain the token from deviating its pre-trained representation, we introduce a

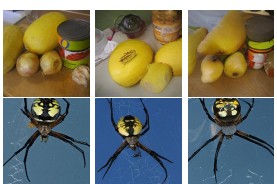

Figure 5: **Fine-tuning the `[cls]` token**. From left: GT, frozen, finetuned.

Table 1: State-of-the-art comparison between tokenizers for reconstruction and class-conditional ImageNet generation. † metrics sourced from (Bachmann et al., 2025).

| Tokenizer | # tokens | global | continuous | rFID | gFID |
|---|---|---|---|---|---|
| LDM (Rombach et al., 2022) | 32x32 | ✗ | ✓ | 0.90 | 7.76 |
| LlamaGen† (Sun et al., 2024) | 16x16 | ✓ | ✗ | 2.19 | 3.06 |
| TiTok-L† (Yu et al., 2024a) | 32 | ✓ | ✗ | 2.21 | 2.77 |
| TiTok-B† (Yu et al., 2024a) | 64 | ✓ | ✗ | 1.70 | 2.48 |
| TiTok-S† (Yu et al., 2024a) | 128 | ✓ | ✗ | 1.71 | 1.97 |
| FlexTok† `d12-d12` (Bachmann et al., 2025) | 1-256 | ✓ | ✗ | 4.20 | 3.83 |
| FlexTok† `d18-d18` (Bachmann et al., 2025) | 1-256 | ✓ | ✗ | 1.61 | 2.02 |
| FlexTok† `d18-d28` (Bachmann et al., 2025) | 1-256 | ✓ | ✗ | 1.45 | 1.86 |
| **RepTok** (*ours*) | 1 | ✓ | ✓ | 1.85 | 1.88 |

cosine-similarity alignment term

$$\mathcal{L}_{\cos}(x) = \lambda(1 - \cos(z, z_{\text{frozen}})) \qquad z_{\text{frozen}} = \mathcal{E}_{\text{frozen}}(x), \ z = \mathcal{E}(x), \tag{3}$$

where $z_{\text{frozen}}$ is the token output from the frozen SSL model, $z$ is the fine-tuned counterpart, and $\lambda$ explicitly controls the allowed deviation. Reducing $\lambda$ relaxes the constraint; increasing it restricts the token more tightly to its source. With this alignment mechanism, we retain the well-behaved SSL latent space for later generative modeling, while additionally enriching the token with the additional information the generative decoder needs to faithfully reconstruct. We observe that incorporating the cosine similarity loss prevents the embedding from drifting away from the well-regularized latent space, also under extended training, as illustrated in Figure 9. We directly condition the generative decoder on those representations and focus on preserving their structured properties while injecting additional information to enable both faithful reconstruction and generative abilities.

### 3.3 SINGLE TOKEN GENERATION FOR IMAGE SYNTHESIS

Since RepTok projects images into a continuous latent space $z$ (typically in $\mathbb{R}^{1 \times 768}$), it becomes feasible to model and sample from this space using a separate generative model $\mathcal{G}$. To this end, we again employ flow matching (Lipman et al., 2023) for latent space generation. We discover that utilizing a frozen SSL model, with only the class token finetuned, provides an effective alternative regularization mechanism to the conventional approaches using Kullback-Leibler (KL) divergence (Rombach et al., 2022) or vector quantization (Austin et al., 2021; Yu et al., 2024a; Tian et al., 2024). By preserving the structural properties of the learned feature space, the frozen encoder inherently constrains the latent representations and facilitates the generation process without requiring explicit KL or vector quantization regularization.

**Attention-free ImageNet Generation** Typical diffusion models operate on high-dimensional image or latent spaces consisting of multiple tokens, where capturing global structure and local detail relies on modeling interactions across tokens. This is commonly achieved through attention (Vaswani et al., 2017). While effective, it introduces significant computational overhead. In contrast, when inputs are aggressively compressed into a single token, token-to-token interactions become unnecessary. We show that in this highly compressed regime, generative modeling can be effectively performed using an attention-free, pure MLP-based architecture such as MLP-Mixer (Tolstikhin et al., 2021). Despite its architectural simplicity and lack of self-attention, our MLP-only approach performs remarkably well. This highlights a novel and computationally efficient approach to diffusion modeling, where architectural complexity is shifted to the pre-trained compression stage without sacrificing flexibility or generality. For text-to-image synthesis, we still use attention for text conditioning, but the compactness of our latent space keeps the associated cost minimal. In particular, because the number of tokens in our latent space is small, the quadratic scaling of attention remains inexpensive.

## 4 EXPERIMENTS

We evaluate RepTok on class-conditional ImageNet-1k (Deng et al., 2009) and show the scalability of our approach on text-to-image (T2I) generation. We evaluate reconstruction performance with reconstruction FID (*rFID*), PSNR, SSIM, and LPIPS, and generation performance with generation

Table 2: FID comparison on the ImageNet $256 \times 256$ benchmark, including parameter counts and training FLOPs. *Stage 1* refers to the training of the generative decoder, while *Stage 2* corresponds to the main generative model training. As all models rely on the SD-VAE and REPA uses DINOv2 as well, we exclude these shared pre-training costs from FLOP estimates.

| Model | FID | Stage 1 | Stage 2 | | | | |
| | | PFlops | Train Steps | Params (M) | GFlops/Iter | PFlops | Total PFlops |
|---|---|---|---|---|---|---|---|
| DiT-XL/2 | 19.5 | – | 400K | 675 | 118.6 | 12.1K | 12.1K |
| +REPA | 12.3 | – | 400K | 675 | 140.5 | 14.4K | 14.4K |
| SiT-L/2 | 18.8 | – | 400K | 458 | 77.5 | 7.9K | 7.9K |
| +REPA | 9.7 | – | 400K | 458 | 99.4 | 10.2K | 10.2K |
| SiT-XL/2 | 17.2 | – | 400K | 675 | 118.6 | 12.1K | 12.1K |
| +REPA | 7.9 | – | 400K | 675 | 140.5 | 14.4K | 14.4K |
| SiT-XL/2 | 8.3 | – | 7M | 675 | 118.6 | 212.5K | 212.5K |
| +CFG=1.5 | 2.06 | – | 7M | 675 | 118.6 | 212.5K | 212.5K |
| +REPA | 5.9 | – | 4M | 675 | 140.5 | 143.9K | 143.9K |
| +REPA, CFG=1.5 | 1.42 | – | 4M | 675 | 140.5 | 143.9K | 143.9K |
| RepTok | 5.4 | 30.4K | 100K | 276 | 23.0 | 0.6K | 31.0K |
| RepTok | 3.4 | 30.4K | 700K | 276 | 23.0 | 4.1K | 34.5K |
| +CFG=1.5 | 3.22 | 30.4K | 700K | 276 | 23.0 | 4.1K | 34.5K |
| RepTok-L | 2.06 | 30.4K | 460K | 516 | 25.0 | 11.7K | 42.1K |
| +CFG=1.5 | 1.88 | 30.4K | 460K | 516 | 25.0 | 11.7K | 42.1K |

FID (*gFID*), consistent with prior work (Bachmann et al., 2025; Yu et al., 2024a). All models operate at $256^2$ resolution; implementation and training details are provided in the Appendix.

## 4.1 CLASS-CONDITIONAL GENERATION

We jointly train the SSL encoder (only the `[cls]` token parameters are trainable) and a generative flow matching-decoder for reconstruction in a first stage. We use DINOv2 (Oquab et al., 2024) as our SSL encoder, but show in Section 4.3 that our method also generalizes to other SSL methods. For latent space synthesis, we train a lightweight, attention-free generator (MLP-Mixer) over the continuous `[cls]` token, where we encode images using the previously trained SSL encoder model. We inject class information by concatenating a learned class embedding, which we randomly drop during training to enable classifier-free guidance (Ho & Salimans, 2021). We apply CFG in a limited interval $t \in [0.3, 0.9]$ following Kynkäänniemi et al. (2024).

**Quantitative Comparison**  Table 2 compares our method against recent, state-of-the-art transformer-based generative models on ImageNet $256 \times 256$. For each model, we report the FID score, number of training iterations, parameter count, per-iteration compute in GFLOPs, and the resulting total training compute in Peta-FLOPs. FLOPs are estimated from a single forward pass (batch size 1), and scaled linearly with the effective batch size and the number of training steps; we follow the convention of counting only the forward pass. Our model achieves highly competitive FID scores while re-

Table 3: Reconstruction performance on ImageNet $256^2$.

| | FID@50K ↓ | PSNR ↑ |
|---|---|---|
| RCG | 3.20 | 9.31 |
| *Ours* | **1.85** | **14.94** |

quiring significantly less total compute than other baselines such as DiT and SiT. We note that classifier-free guidance (CFG) yields only limited improvements in our setting, a phenomenon also reported by RCG. Table 1 compares RepTok with spatial and 1D tokenizers for both reconstruction and class-conditional generation on ImageNet. Despite using just *one* continuous token, RepTok matches or even outperforms several spatial and non-spatial baselines in rFID while remaining competitive on gFID relative to recent discrete tokenizers. Additional results in Table 3 compare RepTok to RCG (Li et al., 2024), a method which relies on purely semantic codes. RepTok achieves significantly higher PSNR and lower FID, indicating that our continuous token preserves more information than pure semantics and delivers stronger performance across both perceptual and pixel-level metrics.

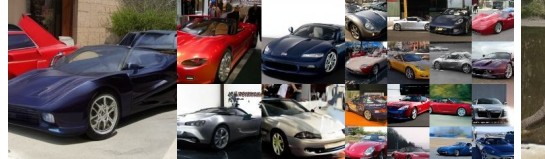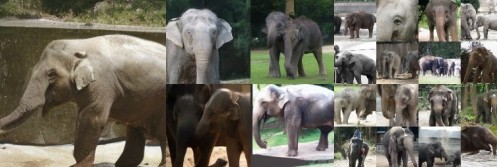

Figure 6: Uncurated MLP-Mixer ImageNet generations (CFG=3.5). More samples in the Appendix.

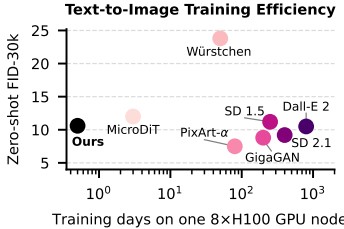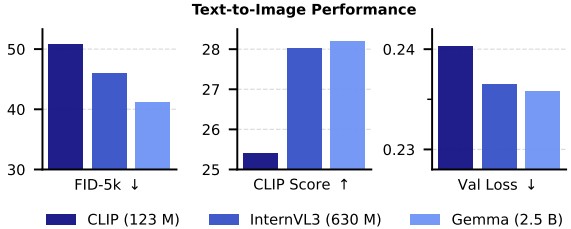

Figure 7: *Left:* Training days vs gFID, zero-shot evaluation on MS-COCO (Lin et al., 2014). Data sourced from MicroDiT (Sehwag et al., 2024). *Right:* Scaling the frozen language backbones results in improved performance. Language models: CLIP, InternVL, and Gemma-2B.

**Efficiency**   We measure training compute in floating point operations (FLOPs). In the single-token latent space, token-to-token interactions are unnecessary. We therefore adopt a pure MLP-Mixer as the latent space generator model. The combination of representing an image with a single token and the MLP-only architecture reduces training FLOPs by an order of magnitude compared to attention-based diffusion in latent space, as shown in Figure 1. Despite a comparable number of parameters across both models, our approach still achieves a substantially lower computational footprint, requiring only 1.7% of the FLOPs consumed by SiT (Ma et al., 2024). Our overall FLOPs remain significantly lower, also when accounting for the inference cost of the corresponding first-stage encoder.

**Qualitative Comparison**   Figure 3 shows high-fidelity reconstructions from a single token on ImageNet validation images and strong out-of-distribution reconstructions on MS-COCO (Lin et al., 2014), despite training only on ImageNet. Figure 6 presents class-conditional samples; despite the simple architecture and low compute budget, quality remains competitive with attention-based image generation models. We provide more uncurated, qualitative samples in the Appendix.

**Latent Space Interpolation**   A key advantage of self-supervised encoders is the smoothness of their latent spaces, yielding a geometry well-suited for generation. Figure 4 shows that our training preserves this property, where we linearly interpolate between latent representations, which produces gradual transitions in both high-level semantics and low-level visual details. We observe continuous changes in object shape, size, emergence, and rotation (see more samples in the Appendix).

### 4.2   ENABLING REPTOK FOR T2I GENERATION

We scale RepTok to text-to-image generation using 120M image–text pairs from COYO (Byeon et al., 2022), recaptioned using InternVL3-1B (Zhu et al., 2025). We first train the language-agnostic

Table 4: Our approach generalizes to other self-supervised encoders. We compare 10k FID on class-conditional ImageNet (Deng et al., 2009).

| SSL method | rFID ↓ | PSNR ↑ | SSIM ↑ | LPIPS ↓ | gFID ↓ |
|---|---|---|---|---|---|
| w/o prior | 13.99 | 19.64 | 47.19 | 0.23 | 128.54 |
| CLIP (Radford et al., 2021) | 13.66 | 14.24 | 31.69 | 0.44 | 30.56 |
| MAE (He et al., 2022) | 9.09 | 13.79 | 30.28 | 0.45 | 28.48 |
| DINOv2 (Oquab et al., 2024) | 7.95 | 14.94 | 33.26 | 0.41 | 20.75 |

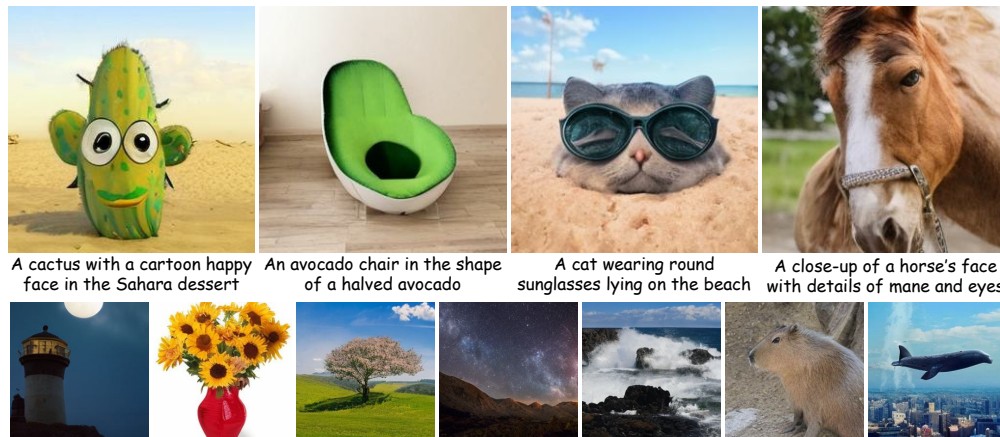

Figure 8: RepTok text-to-image results with a transformer-based latent space model (CFG scale 3.5).

encoder-decoder using DINOv2 as our SSL encoder and a Flow Matching transformer as decoder. During generative model training, we concatenate four learnable tokens with the noisy `[cls]` token from the SSL encoder and apply cross-attention to the frozen outputs of the language model. Similar to prior work, we evaluate our method on the COCO validation set (Lin et al., 2014). We report FID, CLIP Score (Hessel et al., 2021), as well as validation loss, as (Esser et al., 2024; Polyak et al., 2024) found that it correlates with human evaluations.

**Quantitative Results**    Figure 7 (*left*) shows that our method achieves substantially lower training cost than prior text-to-image models while maintaining competitive zero-shot FID. Since the language backbone is frozen and only provides conditioning, it can be scaled independently without impacting the training cost of the generative model. Figure 7 (*right*) shows the performance for language backbones with increasing scale: CLIP (Radford et al., 2021), InternVL (Zhu et al., 2025), and Gemma-2B (Team-Gemma et al., 2024). Larger language models consistently improve performance across all metrics. All results are obtained after 200k training iterations with a batch size of 256.

**Qualitative Results**    Figure 8 shows qualitative text-to-image results. Our model is able to produce realistic images after only 200k training iterations. Despite the short training time ($< 20$ hours on $4\times$ A100 GPUs), the generations capture fine details and adhere closely to the prompt. This highlights the efficiency and scalability of RepTok for text-to-image synthesis. Interestingly, we observe that the SSL encoder and generative decoder trained exclusively on ImageNet can already be repurposed for text-to-image generation. We show qualitative samples and discuss this further in the Appendix.

### 4.3 ABLATIONS

**Generalization to other SSL methods**    Our method generalizes to a number of self-supervised vision encoders, as shown in Table 4. While the main results are based on DINOv2, we observe similarly strong reconstruction quality and generative performance when using alternative SSL methods such as MAE and CLIP. In contrast, when using a randomly initialized encoder with no prior information, the generative decoder loss enforces a strong pixel-wise reconstruction but leaves the resulting latent space completely unstructured and hard to capture for the generative model, as reflected in the high generation FID. A semantic prior enforces a geometry in which semantically similar images are drawn together and dissimilar images are pushed apart. This naturally induces smooth, low-dimensional manifolds which promotes stable generations.

**Cosine Similarity Loss**    We introduced a cosine similarity loss in Equation (3) that incentivizes the semantic token to remain close to the SSL encoder's original to preserve the beneficial properties of the pre-trained space. Here, similar to previous work (Yao et al., 2025; Tschannen et al., 2024), we observe a trade-off between generation and reconstruction, visualized in Figure 9. Stronger regularization improves the generative performance (gFID), but at the cost of reduced pixel-wise reconstruction (PSNR). Mild regularization significantly improves the generative quality, indicating a

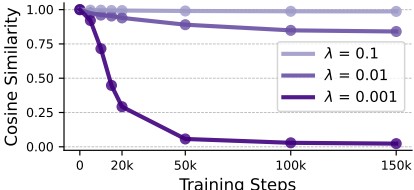 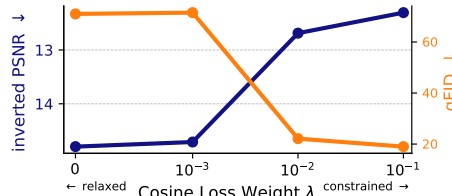

Figure 9: The parameter $\lambda$ of the cosine similarity loss in Equation (3) allows us to trade off between pixel-wise reconstruction and generation capabilities. Relaxed constraints (low $\lambda$) improve pixel-wise reconstruction (PSNR in right plot), but result in poor generation capabilities (gFID in right plot).

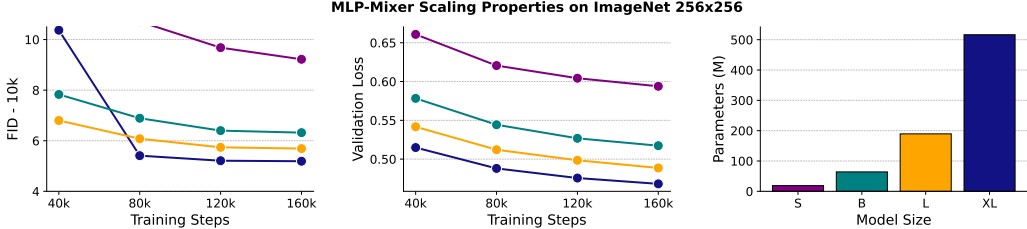

Figure 10: Scaling analysis of the MLP Mixer (Tolstikhin et al., 2021). We observe that the model scales: larger variants consistently yield improved performance, and continued training further improves results. The largest XL model evaluated contains 516M parameters.

better latent space for generation, while minimally degrading reconstruction quality. $\lambda$ allows us to balance between preserving high-level semantic content and reconstructing low-level visual details.

**Scaling Analysis of MLP-Mixer**  Since parameter scaling is a key feature of Transformers, we further evaluate whether this also holds for our latent space MLP-Mixer architecture (Tolstikhin et al., 2021). In Figure 10 we show that the MLP Mixer also scales with parameter size and FLOPs.

**Inference Speed**  We report wall-clock inference time for each component in Table 5. Our MLP-Mixer typically saturates at 40 NFEs, while the SiT-based decoder saturates at 25 NFEs for reconstruction quality (Figure S3).  In comparison, SiT

Table 5: Inference speed comparison.

| Model | NFE | GFLOPs/NFE | Time/NFE (ms) | Total Time (s) |
|---|---|---|---|---|
| SiT-XL | 64 | 118.6 | 38.0 | 2.43 |
| **Ours** | | | | |
| MLP-Mixer | 40 | 3.1 | 6.8 | 0.27 |
| Decoder (SiT-XL) | 25 | 118.6 | 38.0 | 0.95 |

reports saturation at 64 NFEs for the ODE solver (and 256 for SDE). Our overall inference time is dominated by the generative decoder, while the MLP-Mixer generator adds minor overhead.

## 5  CONCLUSION

In this work, we introduced RepTok, a framework that adapts self-supervised representations into a compact latent space for generative modeling. By fine-tuning only the class token of an SSL encoder and regularizing it with a cosine-similarity loss, we obtain a single continuous token that retains the smooth geometry of the original space while enriching it with reconstruction-relevant information. Coupled with a generative decoder trained via flow matching, this setup enables faithful reconstructions and efficient image synthesis without reliance on costly attention mechanisms or auxiliary losses. Our experiments demonstrate that this single-token formulation achieves competitive results in class-conditional generation at a fraction of the computational cost. We further show that RepTok scales to more complex text-to-image settings. Overall, these findings highlight the potential of leveraging SSL representations themselves to build lightweight but effective generative models.

ACKNOWLEDGMENTS

We would like to thank Nick Stracke for helpful discussions and feedback. This project has been supported by the bidt project KLIMA-MEMES, the German Federal Ministry for Economic Affairs and Climate Action within the project "NXT GEN AI METHODS – Generative Methoden für Perzeption, Prädiktion und Planung", the project "GeniusRobot" (01IS24083) funded by the Federal Ministry of Research, Technology and Space (BMFTR), and the Horizon Europe project ELLIOT (GA No. 101214398). The authors gratefully acknowledge the Gauss Center for Supercomputing for providing compute through the NIC on JUWELS/JUPITER at JSC and the HPC resources supplied by the NHR@FAU Erlangen.

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

SUPPLEMENTARY MATERIAL: ADAPTING SELF-SUPERVISED
REPRESENTATIONS AS A LATENT SPACE FOR EFFICIENT GENERATION

## A  IMPLEMENTATION DETAILS

**Generative Decoder**    Our generative decoder is implemented as a DiT-XL/2 (Peebles & Xie, 2023) and trained for one million steps with a learning rate of $10^{-4}$ using the AdamW (Loshchilov & Hutter, 2019) optimizer, a linear warm-up of 2000 steps and a global batch size of 512 on 8 H100 GPUs. Our implementation uses RoPE (Su et al., 2023; Crowson et al., 2024), RMSNorm (Zhang & Sennrich, 2019) and SwiGLU (Shazeer, 2020) activation functions, as we find that these modifications improve the stability and performance of our generative decoder. We concatenate the SSL embedding to the decoder patch tokens and apply full self-attention over all tokens.

**MLP Mixer**    We adopt a standard MLP-Mixer (Tolstikhin et al., 2021) architecture, where all conditioning information: CLIP text embeddings for text-to-image (T2I) generation and class tokens for class-conditional image generation is concatenated with the noisy image token and passed through the model. Our implementation follows the configuration provided by the *lucidrains* [1] GitHub repository, with a hidden dimension of 1280, a depth of 28 layers, an expansion factor of 4 for the channel MLP, and 2 for the token MLP.

## B  ADDITIONAL RESULTS

**Qualitative examples per token type.**    As discussed in the main paper, DINOv2 (Oquab et al., 2024) offers two different types of tokens (besides patch tokens). First, the standard `[cls]` token and additionally a set of register tokens (Darcet et al., 2024). In Figure S4 we provide a qualitative comparison of the differences in outcome between these two token types. We keep the SSL backbone frozen and only train our generative decoder. We can observe that the `[reg]` token contains more knowledge about appearance, location, and object orientation compared to the `[cls]` token. However, none of the approaches gives proper pixel-wise reconstructions, again highlighting the need to integrate further information from the SSL encoder.

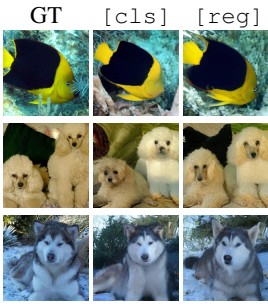

Figure S4: **`[cls]` vs `[reg]`** qualitative comparison.

**Performance vs test-time compute.**    Figure S3 shows the number of function evaluations (NFE) vs reconstruction FID (rFID) on the ImageNet Deng et al. (2009) validation dataset. Performance improves with increasing number of function evaluations (NFE), but saturates around 20. We hypothesize that the strong conditioning signal from the generative decoder reduces the need for additional refinement steps.

**Token type**    DINOv2 (Oquab et al., 2024) provides access to both a `[cls]` token and a set of register tokens. We compare their usefulness as latent representations for our generative decoder in Table 6. Using a frozen `[cls]` token results in strong reconstruction FID, indicating good semantic alignment, but yields low pixel-level scores such as PSNR and SSIM. In contrast, the register token captures more fine-grained visual details, improving pixel-wise reconstruction quality. This suggests that while the `[cls]` token emphasizes semantic content, the register token retains more low-level and regional information.

Table 6: **Ablation of token type.** Conditioning on DINOv2's (Oquab et al., 2024) register tokens improves pixel-wise metrics, indicating stronger local information.

| Token | rFID ↓ | PSNR ↑ | SSIM ↑ | LPIPS ↓ |
|-------|--------|--------|--------|---------|
| `[reg]` | 14.90 | **12.85** | **29.07** | **0.52** |
| `[cls]` | **14.13** | 12.59 | 28.41 | 0.54 |

---

[1] https://github.com/lucidrains/mlp-mixer-pytorch

Image A ⟵——————————— Interpolation ——————————⟶ Image B

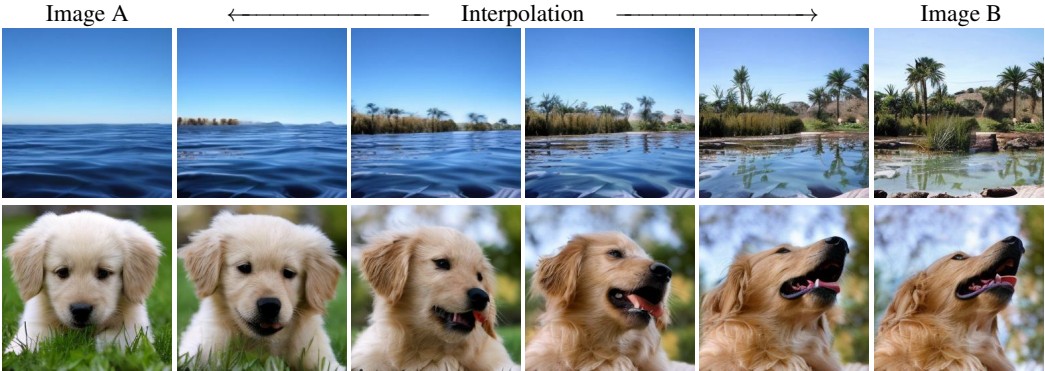

Figure S1: **More single token latent space interpolation results.** We observe smooth transitions not only in semantic content but also in object spatial configuration, and especially in object rotation (see dog).

Image A ⟵——————————— Interpolation ——————————⟶ Image B

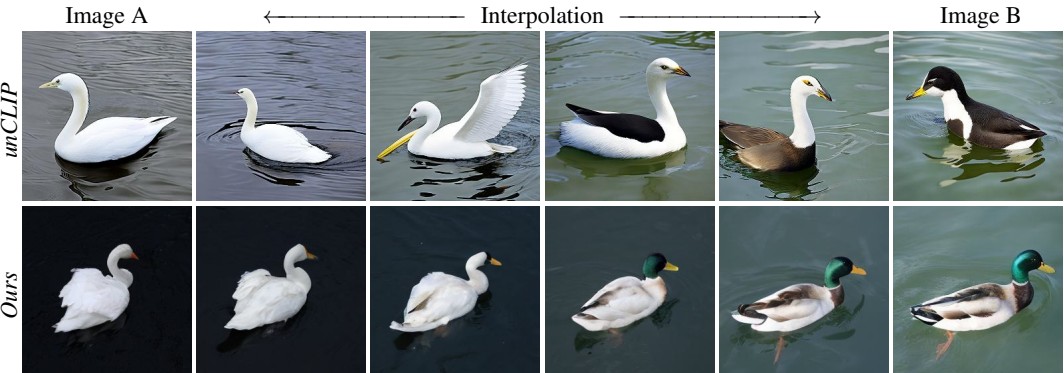

Figure S2: **Qualitative interpolation comparison to _unCLIP_ Ramesh et al. (2022); Rombach et al. (2022)** The results show that representations from unCLIP primarily capture semantic information and lack low-level detail, leading to less coherent transitions. In contrast, our approach preserves both semantic and structural continuity, enabling visually consistent interpolations. We use the pretrained Stable Diffusion 2.1 unCLIP checkpoint.

**Scaling the Decoder to 512px** We further fine-tune our encoder–decoder model for 100K iterations at $512^2$ resolution, starting from weights pre-trained at $256^2$. The model successfully adapts to the higher-resolution setting as visualized in Figure S6.

**More qualitative samples** We provide additional qualitative results to further illustrate the capabilities of our model: text-conditional generations are shown in Figure S10, and uncurated class-conditional ImageNet generations in Figure S12.

**Additional Tokens** Some SSL encoders, such as DINOv2, provide additional global register tokens beyond the [cls] token. Incorporating these tokens increases latent capacity and improves reconstruction quality (see Figure S7, left). However, these tokens are typically unregularized and therefore do not inherit the favorable semantic and well-structured properties that our approach relies on. This also shows in the worse generative performance (gFID in Figure S7, right). Moreover, using multiple global tokens requires SSL models that have additional non-spatial tokens, as there is no straightforward way to repurpose spatial tokens into meaningful global representations. Given our focus on efficiency and structured single-token representations, we therefore restrict our method to the [cls] token, though multi-token extensions remain an interesting direction for future work.

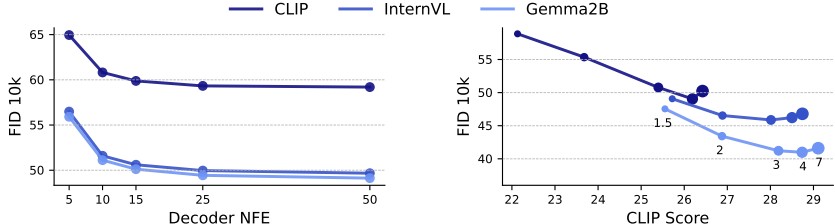

Figure S3: **Effect of decoder inference steps** (left) and **effect of CFG scales** (right). We evaluate both on the COCO validation set Lin et al. (2014). More decoder inference steps yield better decoding results. CLIP score rises with larger CFG scales, while FID improves only within a moderate range.

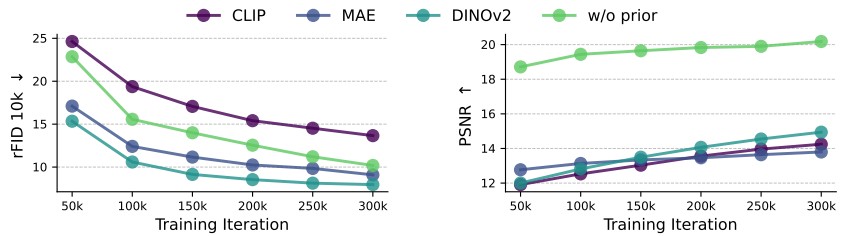

Figure S5: **Comparing SSL priors over training steps.** Our approach generalizes to different self-supervised methods. While the unregularized model without prior knowledge shows remarkable pixel-wise reconstruction, the latent space is not amenable for generation (see Table 4 and Figure S9).

## C    LIMITATIONS

While our single-token representation enables highly efficient generation and significant compute savings, it may limit expressiveness in capturing fine-grained details, particularly for complex or high-resolution scenes. Extending our approach to support richer multi-token representations while preserving efficiency is an interesting direction for future work. While our experiments demonstrate that the single-token embedding preserves certain low-level spatial structures, achieving fine-grained control over object location and scene composition remains an open challenge.

**Reconstruction-Generation trade-off**    Another limitation of our method lies in the trade-off imposed by cosine similarity regularization. While stronger regularization enhances the smoothness and structure of the latent space, which is crucial for stable generative modeling, it can also suppress low-level detail, leading to degraded pixel-wise reconstructions. This trade-off may limit the applicability of our approach in scenarios where very high visual reconstruction fidelity is critical.

**Unleashing T2I for ImageNet-Pretrained Autoencoder**    We investigate the capabilities of our Image-trained encoder-decoder framework. Figure S11 shows qualitative text-to-image samples. Despite being trained exclusively on ImageNet, the latent space does not overfit and shows strong

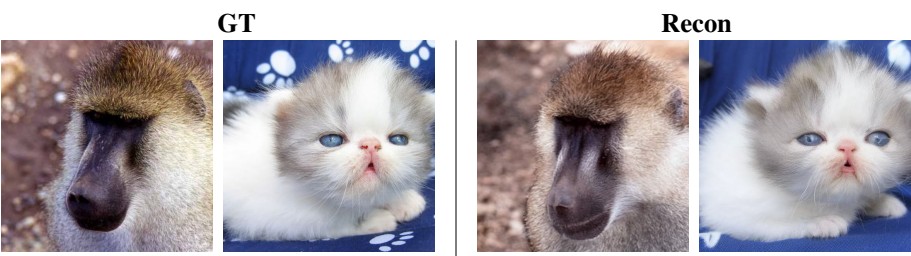

Figure S6: Fine-tuned encoder-decoder model on the higher $512^2$ resolution.

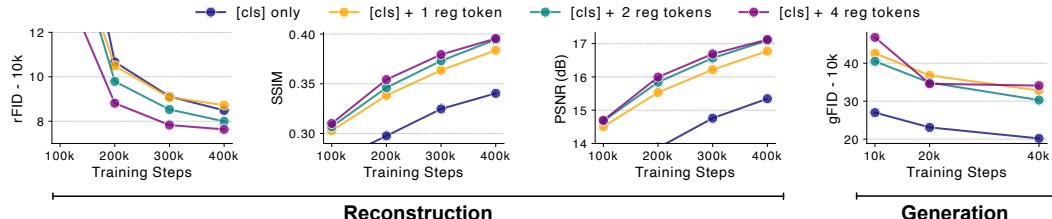

Figure S7: **Token number ablation.** Increasing the number of tokens in form of additional register tokens from DINOv2 Darcet et al. (2024) improves reconstruction quality. However, since register tokens do not have the favorable SSL properties, their space is not amenable for generation.

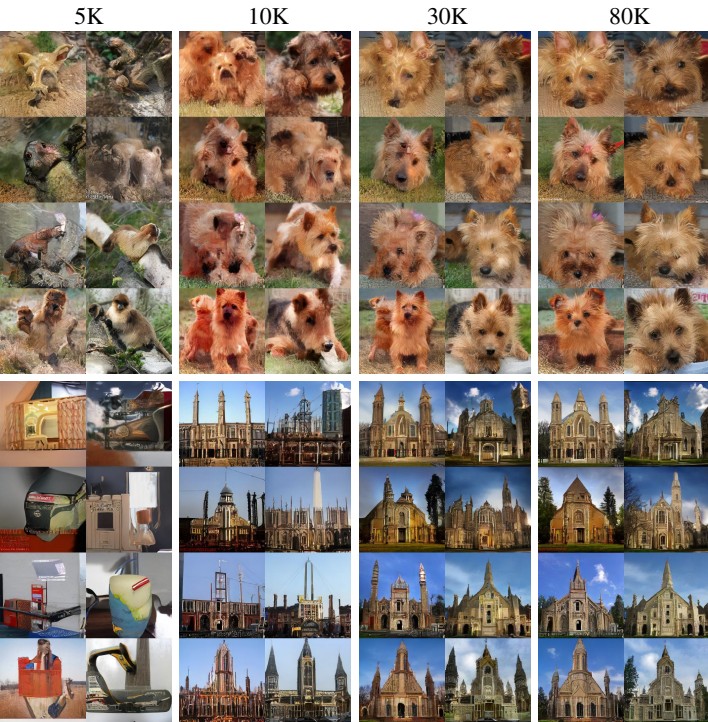

Figure S8: **Uncurated** class-conditional ImageNet generation results over training iterations (5k, 10k, 30k, and 80k). Note that our model produces good results as early as 30k training steps.

generalization, generating diverse and high-quality images that extend well beyond the ImageNet manifold. Although the model generates plausible images, we find it struggles with compositional prompts that require placing multiple objects within a scene (e.g., a cat and a dog side-by-side). This limitation is expected, since the object-centric bias of ImageNet offers little exposure to multi-object scenes. However, finetuning our encoder on more diverse data alleviates this issue and enables the generation of multi-object content.

**Reconstruction**          **Generation**

GT          Random          Ours          Random          Ours

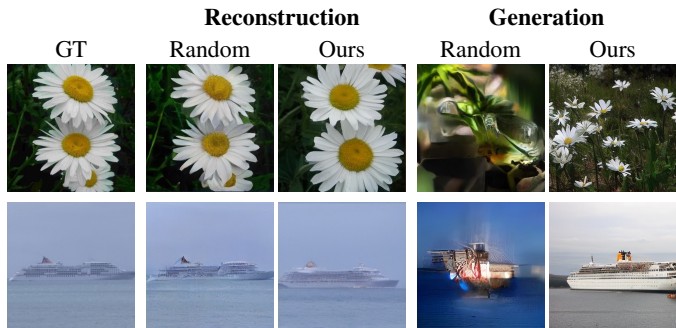

Figure S9: Qualitative comparison between a randomly-initialized encoder and ours. Generation refers to class-conditional samples with the same class as the corresponding GT image. While random initialization achieves stronger pixel-level reconstruction, it lacks the structured priors of pre-trained self-supervised encoders, resulting in poor generative performance. In contrast, our method balances reconstruction and generation.

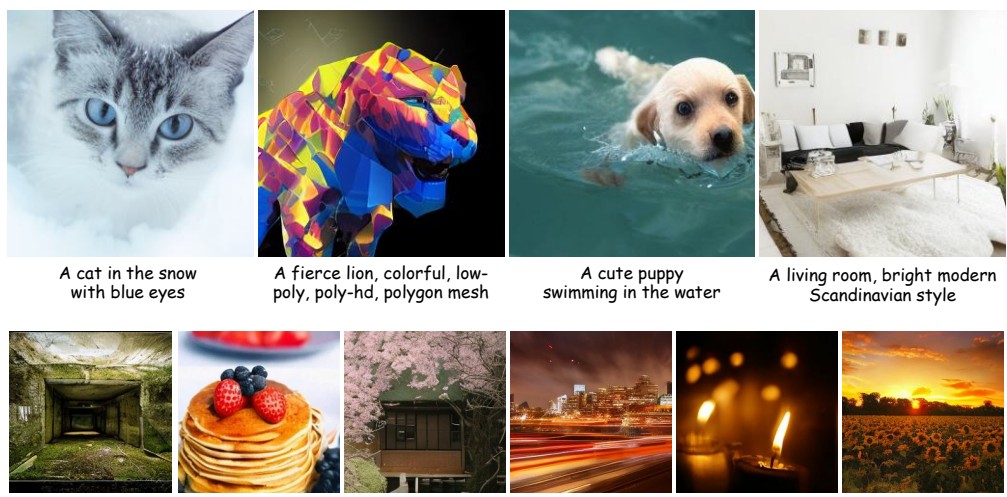

Figure S10: Additional text-to-image generation results with a CFG scale of 7.5 and RepTok encoder-decoder trained on the COYO dataset.

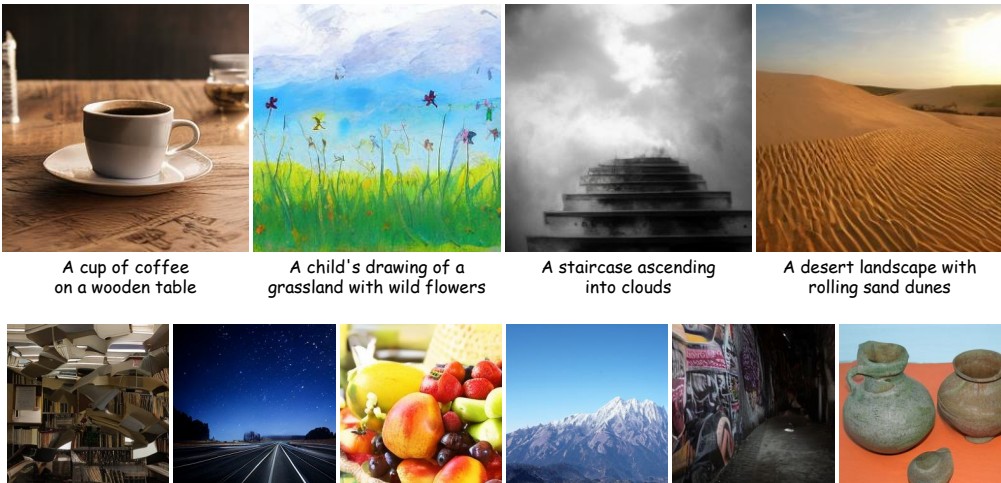

Figure S11: T2I generation results (CFG scale 3.5), using RepTok solely trained on ImageNet data with a latent space transformer. The autoencoder also transfers effectively to T2I tasks, producing visually compelling results.

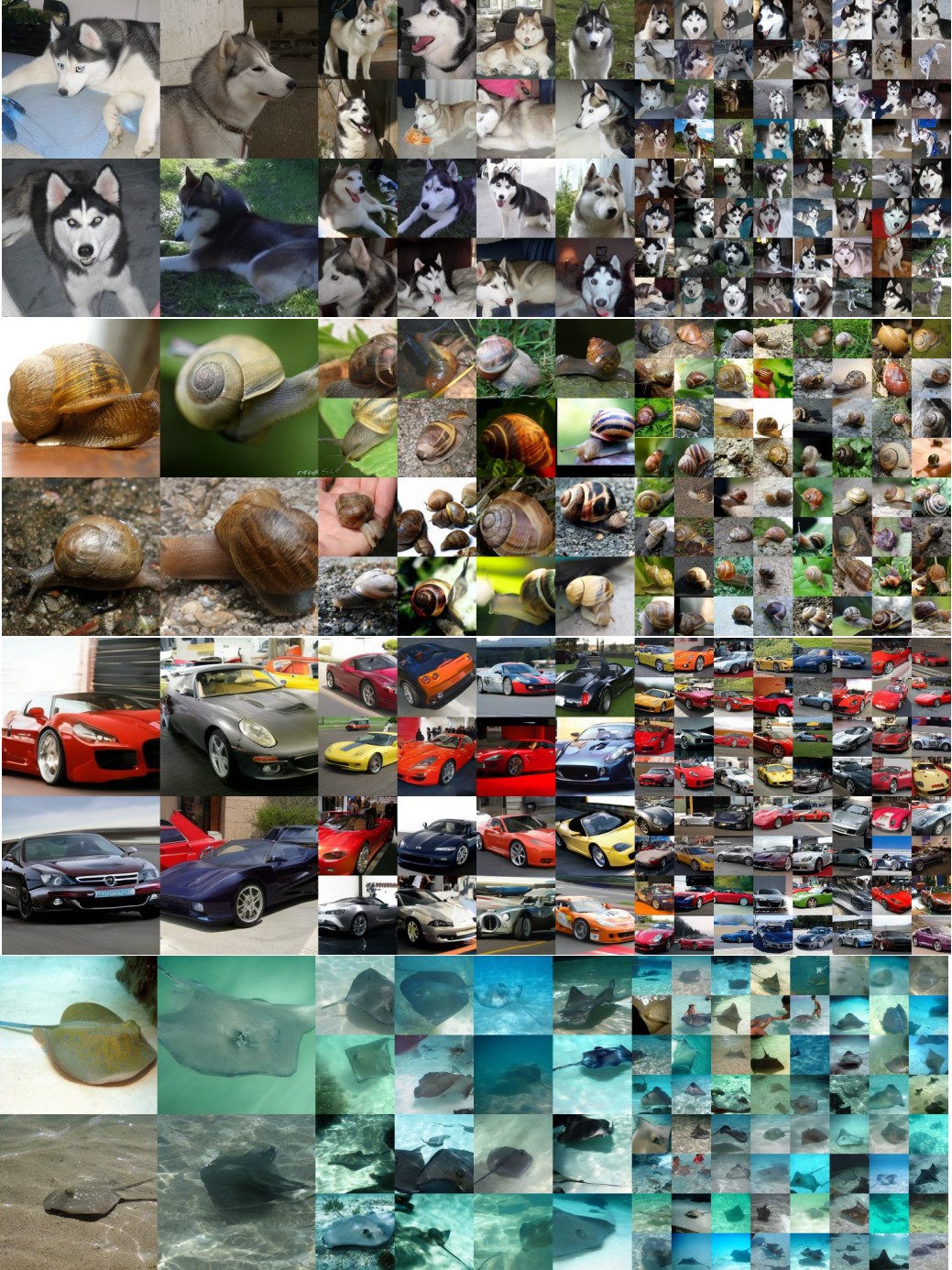

Figure S12: **Uncurated** class-conditional generation results of RepTok with CFG scale of 3.5.

