# OpenReview forum: "Adapting Self-Supervised Representations as a Latent Space for Efficient Generation"
_ICLR.cc/2026/Conference — ICLR 2026 Poster_

### Official Review · Reviewer_kicm · 2025-10-16

**Soundness:** 2
**Presentation:** 3
**Contribution:** 2
**Rating:** 4
**Confidence:** 3

**Summary:**

This paper adapts pre-trained self-supervised representations into a single continuous latent token space and trains a lightweight latent generator with a generative decoder. This idea is potentially impactful for compute-efficient generation. However, at the reported compute and model sizes, the performance falls short of the current SOTA, which lacks a matched-budget comparison. Also, the Flops reported in Table 3 appears to exclude the generative decoder, making the efficiency claims for end-to-end generation unclear. The paper also lacks a scaling study of compute resource and paramters which is an important property.

**Strengths:**

1. It is a noval approach that uses a single continuous latent token adapted from self-supervised representations with a lightweight latent generator.
2. RepTok shows flexibility as it extents to text-to-image synthesis.
3. The paper is generally clear and well organized.

**Weaknesses:**

1. Although the compute and model sizes of latent generator is smaller, the gFID in Table 2 and FID in Table 3 are not competitve with SOTA. It remains unclear how RepTok performs under matched evaluation settings against baselines.
2. Table 3 appears to report their compute resources and parameters only for the MLP Mixer, excluding the stage-one Generative Decoder. This undermines the fairness of the compute comparison and weakens the efficiency claim.
2. The paper does not provide a scaling study for the MLP-Mixer generator (e.g., FID vs. parameters/train steps). Table 3 reports only a single 276M model, whereas the only scaling evidence shown concerns the frozen language backbones in Figure 7. Since parameter scaling is a key feature of DiT/SiT, reporting two training-step settings is insufficient to establish scaling behavior.
3. There are many instances of incorrect citation formatting: using author-in-text citations (\citet{...}) instead of the required parenthetical author-year style citations (\citep{...}).

**Questions:**

1. What are the total compute resources and paramters combining the Generative Decoder with the MLP Mixer? Please report FID, GFlops/Iter, total PFlops in Table 3 under matched evaluation settings and include the stage-one generative decoder in compute resources and parameters.
2. Can RepTok exceed SOTA FID when scaled up in Table 3? Please include a scaling analysis of the MLP-Mixer sizes and longer training, reporting FID vs. total PFLOPs/params under matched evaluation settings.

---

> ### Author Response · Authors · 2025-11-21
>
> Dear reviewer kicm,
>
> Thank you for your helpful comments. We will address your concerns below.
>
> ---
>
> ## W1 Increased latent space model capacity
>
> Thank you for bringing this up. With the increased capacity of our latent space model, we now achieve competitive performance with state-of-the-art methods (see updated Tables 2 and 3) while still requiring substantially fewer computational resources, owing to the efficiency of our compact latent space and the scalability of the MLP-Mixer.
>
> ## W2 Stage-one decoder training cost
>
> In the original submission, Table 3 focused on the compute cost and parameter count of the MLP-Mixer component. Following your suggestion, we now report the total compute and parameter counts of our full training pipeline (Generative Decoder + MLP-Mixer) under the same evaluation protocol in Table 3. Our generative decoder is a slightly modified SiT-XL, which is trained for 1M steps, accounting for 30.4k P-FLOPs. Despite this modest compute budget, our model achieves competitive or even stronger FID scores compared to substantially more expensive baselines. Importantly, the first-stage model only needs to be trained once; afterwards, the latent-space model can be scaled independently to further improve performance (see Figure 10). Regarding pretrained components such as DINOv2 and the SD-VAE, we exclude the training cost from our computation estimates, as it is difficult to estimate and is also used broadly across other generative approaches.
>
>
> | Model            | FID  | Stage 1 PFlops | Train Steps | GFlops/Iter | Stage 2 PFlops | Total PFlops |
> |------------------|:------:|--------------:|-------------:|-------------:|---------------:|---------------:|
> | SiT-XL/2         | 17.2 | --             | 400K        | 118.6       | 12.1K          | 12.1K         |
> | + REPA           | 7.9  | --             | 400K        | 140.5       | 14.4K          | 14.4K         |
> | SiT-XL/2         | 8.3  | --             | 7M          | 118.6       | 212.5K         | 212.5K        |
> | + REPA           | 5.9  | --             | 4M          | 140.5       | 143.9K         | 143.9K        |
> | **RepTok** (ours)         | 2.06 | 30.4K          | 460K        | 25.0        | 11.7K          | 42.1K         |
>
>
> ## W3 Scaling of MLP-Mixer
>
> Thank you for bringing this up. While our primary emphasis in the paper is on overall efficiency, we agree that understanding how the MLP-Mixer generator scales is important. Motivated by your suggestion, we conducted an additional scaling study and found that increasing the capacity of the MLP-Mixer further improves performance. We now include these results in Figure 10, which demonstrates clear and consistent scaling behavior as model size and training steps increase, following trends similar to those observed in transformer-based generators, such as SiT and DiT.
>
> ## W4 Citation format
>
> Thank you for bringing this to our attention. We have fixed the citation style throughout the paper.

---

> > ### Comment · Reviewer_kicm · 2025-11-24
> > **Thanks for your response**
> >
> > Thank you for your patient and detailed response. You have addressed most of my concerns, and I appreciate the effort you put into the clarification. I have accordingly raised my score in recognition of your improvements.

---

> > > ### Author Response · Authors · 2025-11-28
> > >
> > > We’re glad to hear that we addressed most of your concerns, and appreciate that you raised your score to positive (6) on Nov. 24th. Thank you for your feedback!

---

### Official Review · Reviewer_GhVu · 2025-10-27

**Soundness:** 3
**Presentation:** 3
**Contribution:** 4
**Rating:** 8
**Confidence:** 4

**Summary:**

This paper introduces RepTok, a generative modeling framework that leverages the [cls] token from a pre-trained self-supervised vision transformer as a single-token, continuous latent space useful for image generation tasks.

The paper fines only the [cls] token from a pretrained SSL network and regularizing it with a cosine similarity loss. The authors enable faithful image reconstruction and efficient generative modeling. The approach achieves competitive results on class-conditional ImageNet generation and zero-shot text-to-image synthesis on MS-COCO.

This paper, to my knowledge, is the first single-token encoder paper that achieves reasonably high performance on image genertion tasks

**Strengths:**

(+) novelty: using a single [cls] token to represent the image in the continuous space to my knowledge is novel (and neat), what is better is that it can be minimally fine-tuned from a pretrained SSL network. The paper can encourage more work for compact visual tokenizer that improves both token redundancy and improve tokenization/reconstruction quality

(+) Competitive performance: the performance is good for the first paper with a single-token compression. The experiements are also relatively thorough

(+) Generalization: can use different SSL models, such as DINOv2, MAE, CLIP

**Weaknesses:**

(-) The method requires careful tuning of the cosine similarity loss parameter and careful choices of freeze/unfreeze tokens to balance between reconstruction fidelity and generative quality. This trade-off may be non-trivial in practice and may need tuning for different backbones

(-) A single-token representation is neat, but it will be nice to see the tradeoff between more tokens vs single token.

**Questions:**

n/a

---

> ### Author Response · Authors · 2025-11-21
>
> Dear reviewer GhVu,
>
> Thank you for your helpful comments. We address your concerns below.
>
> ---
>
> ## W1 Tuning of hyperparameters
>
> Our method indeed introduces a cosine similarity weighting parameter that controls the trade-off between reconstruction fidelity and regularization. This type of balance is inherent to latent-space generative models: for example, VAEs rely on a $\beta$-weighted KL term to regulate the same trade-off. Across all experiments and backbones considered in the paper, we empirically find that setting the weighting parameter to $\lambda = 0.01$ consistently yields stable training and competitive results without further adjustments.
>
> ## W2 Trade-off between single vs multiple tokens
> Thank you for the suggestion. In principle, DINOv2 provides additional register tokens that could potentially serve as extra latent tokens beyond the pooled representation. However, this option is not universally available across SSL models (e.g., CLIP does not offer such tokens).
> Meanwhile, we notice that a concurrent work, Representation Autoencoder (RAE), explores the opposite end of this spectrum by using the full set of DINO spatial tokens as the latent space. This highlights an inherent trade-off: using more (spatial) tokens can improve reconstruction fidelity, but it comes at a substantially higher computational cost during both training and inference. Our single-token design is intended to push the efficiency side of this trade-off to an extreme.
>
> **Update**:
> We further ablate the reconstruction quality with multiple tokens (register tokens from DINOv2) in Figure S5 in the updated supplementary material. We observe that increasing the number of tokens improves reconstruction quality. However, this strategy relies on SSL encoders that have non-spatial (register) tokens. Selecting spatial tokens is non-trivial as they are inherently localized and can hardly capture global information. Note that also register tokens, unlike the class token, are typically non-regularized in the SSL training and therefore do not inherit the favorable properties that our approach relies on, as shown by the generative performance in Figure S5 right. Moreover, increasing the number of tokens also increases computational cost during training and inference. Since the primary focus of this work is efficiency, we stick to the single-token representation which is most efficient, most generalizable, and also works well in practice. Nonetheless, exploring multi-token variants and intermediate trade-offs remains an interesting direction for future research.

---

### Official Review · Reviewer_TyLM · 2025-10-31

**Soundness:** 3
**Presentation:** 3
**Contribution:** 3
**Rating:** 6
**Confidence:** 4

**Summary:**

This paper proposes RepTok, an approach for training image generative models using pre-trained self-supervised learning (SSL) models. During the encoder-decoder training phase, the SSL model is fine-tuned efficiently via a learnable [CLS] token parameter, and a flow matching head is trained. In the image generation training phase, images are generated in the SSL model's latent space using flow matching based on an MLP model. The paper validates the feasibility of using SSL as an encoder and demonstrates impressive performance in both image reconstruction and generation.

**Strengths:**

1.  The paper offers several intriguing insights in its model architecture design. These include the indirect fine-tuning of the pre-trained SSL model via a learnable [CLS] token, the use of cosine similarity as a latent space regularizer, and the employment of an MLP-Mixer as the backbone for the generative stage. I believe these interesting design choices could provide inspiration for subsequent researchers.
2.  The proposed method is notably lightweight. On one hand, the model architecture does not require training an excessive number of parameters; on the other hand, building upon a pre-trained SSL model keeps the training cost for the generative model relatively low, suggesting potential for practical application.
3.  The paper is generally well-written and clear, and the experiments are reasonably comprehensive.

**Weaknesses:**

1.  Although the structure is lightweight, the scaling potential of the method is concerning. Firstly, while the reconstruction results in Figure 3 appear superior to FlexTok, they do not seem to surpass those of the VAE. Furthermore, as shown in Table 3, RepTok's final performance is lower than that of REPA-CFG 1.5. Therefore, I would like the authors to provide an analysis of how this method could be further scaled, including a discussion of performance bottlenecks, and to present additional experimental results demonstrating the method's potential at larger scales.
2.  The authors' proposed method (two-level flow matching) bears formal resemblance to Transition Matching [1]. Furthermore, the idea of replacing VAEs with self-supervised models has also been explored, including in works such as RAE [2] and SVG [3]. While direct comparison with concurrent work is not strictly necessary, I encourage the authors to supplement the discussion with a clarification of the distinctions and a more in-depth comparative analysis.
3.  Since the authors' core claim also includes that "SSL provides a better semantic space than VAE" (not merely lightweight), additional experimental results are needed to substantiate this point, particularly regarding performance aspects. This could include evaluations on avoiding distortions and handling challenging, complex, or compositional prompts.
4.  The citation format is informal; it should be (Ho et al. 2020), not Ho et al. (2020).

ref:

[1] Transition Matching: Scalable and Flexible Generative Modeling

[2] Diffusion Transformers with Representation Autoencoders

[3] Latent Diffusion Model without Variational Autoencoder

**Questions:**

Refer to the Weaknesses section.

---

> ### Author Response · Authors · 2025-11-21
>
> Dear reviewer TyLM,
>
> Thank you for your feedback. We address your concerns below.
>
> ---
>
> ## W1 Scaling potential
>
> Our submission emphasizes efficiency and demonstrates that lightweight architectures can already deliver competitive reconstruction and generative performance. Our intention is not to claim superiority over a full VAE at the current scale, but rather to demonstrate that a simple semantic-token encoder with a compact representation can also achieve strong fidelity, despite being significantly smaller and more efficient.
>
> Regarding scalability, our method consists of three components, each of which offers avenues for scaling:
>
> - **SSL encoder**: The current experiments use DINOv2, but the encoder could naturally be upgraded to stronger SSL models such as DINOv3.
> - **Generative decoder**: Our decoder is transformer-based, and transformers are known to scale effectively with both depth and width.
> - **Latent MLP-Mixer**: We conducted additional experiments examining the scaling behavior of the MLP-Mixer. As shown in the updated version of Figure 10, increasing its capacity consistently improves generative performance (as shown in Table 3), indicating that the latent generator also exhibits meaningful scaling potential.
>
>
>
> ## W2 Comparison to concurrent work
>
> We thank the reviewer for highlighting the connection to Transition Matching (TM) and SSL-based latent model alternatives such as RAE and SVG. Aiding image synthesis with representation priors is a rapidly evolving research area, and we are pleased to see complementary works pushing this direction forward. Importantly, both RAE and SVG were posted on arXiv **after the ICLR submission deadline**, but we are very happy to discuss their relationships to our method in detail.
>
> **Relation to RAE and SVG**
> We appreciate the reviewer's observation that several recent works explore replacing VAEs with SSL-derived latent spaces. While related in spirit, RAE and SVG retain the full spatial grid of SSL features (or align the generative model to those grids), thus operating in a high-dimensional structured latent space.
> Our approach is fundamentally different: RepTok relies solely on the pooled semantic token, discarding all spatial tokens and learning to represent an image with a single compact vector. This yields a much more aggressive compression and a conceptually distinct formulation.
> Additionally, a key difference to SVG is that it is aligned to the SSL representation, whereas we directly use the pooled semantic output as the latent representation itself.
>
> **Transition Matching**
> While we see the conceptual similarity at high level where both techniques involve multi-stage formulations, we argue that the underlying principles differ substantially.
> TM explicitly models a discrete-time stochastic process by learning transition kernels with supervision at each intermediate interpolant. Its core is learning a temporal composition where each step learns the transition between successive latent states.
> In contrast, RepTok focuses more on generating a compact representation of the image without providing direct supervision on the intermediate latent space. Our two-stage structure is not temporal but hierarchical:
> Stage 1 produces a compact semantic latent representation of the image derived directly from a pretrained SSL model.
> Stage 2 "interprets" this latent into pixel space using a generative decoder.
>
> ## W3 Comparison between RepTok and VAE
> VAEs have been extensively and successfully used for generative modeling, and we do not assert that SSL-based latents universally outperform them across all aspects such as distortion avoidance. Our intended claim is more modest and specific: we aim to demonstrate that SSL models provide an alternative and viable single-token latent space for generative modeling. The SSL space offers certain qualitative benefits: a compact representation and well-structured latent space amenable for generation. Rather than using these properties to align the latent space model (e.g. REPA), our method directly exploits them for latent space generative modeling.
>
> ## W4 Citation formatting
> Thank you for bringing this to our attention. We have fixed the citation style throughout the paper.

---

### Official Review · Reviewer_wivx · 2025-11-01

**Soundness:** 4
**Presentation:** 4
**Contribution:** 3
**Rating:** 6
**Confidence:** 3

**Summary:**

This paper presents a method for tokenizing an image into a single continuous token using an SSL-pretrained model. Based on this single token, the authors design a generative decoder and a representation generator to perform reconstruction and generation.

**Strengths:**

1. This paper provides strong evidence that SSL models can provide strong generative information using only a single image token.
2. The corresponding generative decoder and representation generator are well designed.
3. The proposed method shows very high training efficiency.

**Weaknesses:**

1. Lack of representation comparison: As the work is heavily based on the image representations, it would be helpful to show the correspondence between generation quality and representation performance under different [CLS] settings (e.g., frozen or fine-tuned with varying $\lambda$).
2. The authors experimented with different SSL models for the proposed method. How does the performance change when combining multiple models, such as training a lightweight linear projection on top of the concatenated [CLS] tokens (or through other fusion strategies)?
3. The proposed method heavily relies on image SSL models, which may limit its generalizability to video generation or multimodal generation tasks.
4. The paper only conducts experiments at a resolution of 256x256. Using a single token to encode an image may be ineffective for higher-resolution settings.

**Questions:**

As the generation process involves two flow-matching steps, I am curious about the inference speed of the proposed model. Could the authors provide a comparison of inference time?

---

> ### Author Response · Authors · 2025-11-21
>
> Dear reviewer wivx,
>
> Thank you for your feedback. We address your comments below.
>
> ---
>
> ## W1 Correlation between hyperparameter
>
> As shown in Fig. 9 (left), using small values for $\lambda$ leads to a substantial drop in cosine similarity between the predicted token and the original SSL embedding. This indicates that the latent representation drifts away from the pretrained semantic space, thereby weakening its representational quality and harming generation. In contrast, larger $\lambda$ values preserve the semantic structure but may slightly constrain reconstruction fidelity. We conducted an additional experiment by performing linear probes on the original DINOv2 encoder and on its finetuned variants under different $\lambda$ settings. We find that $\lambda = 0.01$ provides a good balance between maintaining semantic, smooth structure and accurate reconstruction. Notably, while our generative decoder is able to reconstruct images across all $\lambda$ settings, the corresponding linear probes perform considerably worse, indicating that the encoded information remains recoverable but becomes too nonlinear for a simple linear classifier to extract.
>
> | Method                        | ImageNet Linear Probe Accuracy |
> |------------------------------|---------------------------------|
> | DINOv2                       | 84.5                            |
> | RepTok ($\lambda=0.1$)       | 80.4                            |
> | RepTok ($\lambda=0.01$)      | 72.7                            |
> | RepTok ($\lambda=0.001$)     | 17.6                            |
>
> ## W2 Combining multiple SSL methods
>
> In principle, the proposed method does not constrain the choice of SSL backbone, nor does it preclude the combination of multiple pretrained models. However, determining how different representations interact and how complementary they are is a purely empirical question and calls for a systematic investigation that is both an exciting direction for future works while also falling out of scope for the current submission.
>
> ## W3 Generalizability to other modalities
>
> While our experiments in the paper focus on images, we would like to emphasize that the proposed framework is not fundamentally restricted to image SSL models, but open to the world of self-supervised representation learning across other modalities.
> SSL models for text (e.g., BERT, GPT-style masked/next-token pretraining), video (e.g., V-JEPA, VideoMAE, OmniMAE), and other modalities provide structured latent representations similar to those produced by image SSL models. Extending RepTok to these domains is an interesting direction for future work, and we believe the underlying formulation clearly lends itself to such multimodal generalizations.
>
> ## W4 Larger resolution (512px)
>
> While our experiments focus on the $256^2$ setting, extending the method to higher resolutions is conceptually straightforward. Generating larger images requires encoders and decoders of higher capacity, as they must supply additional high-resolution detail on top of the lower-resolution information that our model has already demonstrated it can effectively capture in the main paper.
>
> To illustrate this, we finetune our $256^2$ encoder–decoder model on $512^2$ resolution. As shown in Fig. 11, with 100k additional training iterations, the model adapts to the higher-resolution setting.
>
> ## Q1 Inference speed per stage
>
> To quantitatively assess inference efficiency, we compute the wall-clock inference time per function evaluation for each component of the model. Our MLP-Mixer stage typically uses 40 NFEs, which introduces only a small overhead relative to the SiT-XL decoding stage. The overall inference time is dominated by the generative decoder rather than by our latent generator.
>
> As shown in Fig. S2a (supplementary material), our SiT-based decoder achieves saturation at around 25 NFEs for reconstruction quality. In contrast, the SiT paper reports that the ODE solver saturates at around 64 NFEs, and the SDE solver requires around 256 NFEs to reach optimal performance, indicating that the generative decoding stage seems more inference-efficient than in standard SiT setups, while the additional MLP-Mixer stage does not constitute a significant computational burden.
>
> | Model                  | NFE | GFlops/NFE | Time/NFE (ms) | Total Time (s) |
> |------------------------|----:|-----------:|--------------:|---------------:|
> | SiT-XL                |  64 |      118.6 |          38.0 |            2.4 |
> | **Ours**              |     |            |               |                |
> | └─ MLP-Mixer          |  40 |        3.1 |           6.8 |           0.27 |
> | └─ Decoder (SiT-XL)   |  25 |      118.6 |          38.0 |           0.95 |

---

### Author Response · Authors · 2025-11-21

Dear reviewers,

Thank you for your constructive feedback. We are glad that you found RepTok “efficient” (**wivx, TyLM**) and “flexible” (**GhVu, kicm**), and appreciate that you find our method “well designed” and “intriguing” (**wivx, TyLM**). We are also encouraged that you found the idea novel (**GhVu, kicm**), described the paper as “clear” and “well written” (**TyLM, kicm**), and regarded the ability to recover an image from a single token as particularly promising (**wivx**).

We also appreciate that reviewers emphasized the broader potential of our approach, noting that it may “inspire subsequent researchers” (**TyLM**), offer “practical applications” (**TyLM**), and “encourage more work on compact visual tokenizers” (**GhVu**).

We highlight updates to the paper in $\color{blue}{\textit{blue}}$. In the updated version, we corrected the citation formatting, added a MLP-Mixer scaling experiment (Fig. 10), and included first-stage training cost and large MLP-Mixer FID results (Tab. 3). We updated the generative FID numbers in Tab. 2, added concurrent work to the related-works section, and added a brief description and qualitative results for the model adaptation to $512^2$ resolution.

Below we address each individual comment.

---

### Meta-Review · Area_Chair_Cao9 · 2026-01-10

**Summary:**

This paper presents RepTok, a novel and efficient approach that uses a single self-supervised representation token as a very compact latent for generative modeling. Reviewers consistently highlighted the novelty, clean design, and strong efficiency story. Initial concerns mainly focused on scaling, fairness/clarity of compute comparisons, and how strong the claims are relative to VAE-based methods. After the rebuttal and discussion, reviewers generally agreed that the paper makes a solid and meaningful contribution and should be accepted.

**Reviewer Concerns:**

The rebuttal addressed most of the key concerns, including clearer end-to-end compute accounting, added scaling results, inference efficiency analysis, and better explanation of the reconstruction vs. generation trade-offs. Some broader questions around absolute SOTA performance and whether SSL latents are strictly better than VAEs remain open, but these are mostly about positioning and scope rather than technical soundness, and are not blockers.

**Reviewer Scores:**

The rebuttal went pretty well. i didn't expect much change.

---

### Decision · Program_Chairs · 2026-01-26

Accept (Poster)